# A Rapid On-Line Evaluation (ROLE) Protocol in the Diagnostic Performance Improvement in Endoscopic Ultrasound-Guided Tissue Acquisition for Solid Pancreatic Lesions

**DOI:** 10.3390/diagnostics14060597

**Published:** 2024-03-12

**Authors:** Yunlong Cai, Xiaolong Rao, Jixin Zhang, Guanyi Liu, Yiling Zheng, Taohua Yue, Weidong Nian, Long Rong

**Affiliations:** 1Endoscopy Center, Peking University First Hospital, Beijing 100034, China; cyl030104@163.com (Y.C.); rxl624@163.com (X.R.); guanyiliu@pku.edu.cn (G.L.); duanmo02@126.com (Y.Z.); taohua_yue@bjmu.edu.cn (T.Y.); nwd6609@sina.com (W.N.); 2Pathology Department, Peking University First Hospital, Beijing 100034, China; jixin.zhang@pkufh.cn

**Keywords:** rapid on-line evaluation, conventional rapid on-site evaluation, solid pancreatic lesion, endoscopic ultrasound-guided tissue acquisition, diagnostic performance

## Abstract

We assessed the rapid on-line evaluation (ROLE) protocol as a modification to the conventional rapid on-site evaluation (ROSE) in the diagnostic performance improvement in endoscopic ultrasound-guided tissue acquisition (EUS-TA) for solid pancreatic lesions. This single-center, retrospective study involved consecutive patients with solid pancreatic lesions undergoing EUS-TA at Peking University First Hospital between October 2017 and March 2021. Among 137 patients enrolled, 75 were in the ROLE group and 62 were in the non-ROSE group. The diagnostic yield (97.3% vs. 85.5%, *p* = 0.023), accuracy (94.7% vs. 82.3%, *p* = 0.027), and sensitivity (95.7% vs. 81.1%, *p* = 0.011) were significantly higher in the ROLE group compared to the non-ROSE group. However, specificity, positive predictive value, negative predictive value, and area under the curve (AUC) showed no significant differences (all *p*-values > 0.05). Additionally, there was a noteworthy reduction in the number of needle passes required in the ROLE group compared to the non-ROSE group (two vs. three, *p <* 0.001). In a subgroup analysis, fine needle biopsy (FNB) combined with ROLE demonstrated superior diagnostic accuracy compared to FNB with non-ROSE (100% vs. 93.1%, *p* = 0.025). Compared with the non-ROSE protocol, the ROLE protocol might improve the diagnostic performance of EUS-TA for solid pancreatic lesions, and potentially reduce the number of needle passes requirement.

## 1. Introduction

Endoscopic ultrasound-guided tissue acquisition (EUS-TA), encompassing endoscopic ultrasound-guided fine needle aspiration (EUS-FNA) and biopsy (EUS-FNB), is pivotal for diagnosing pancreatic solid lesions, with studies reporting 69–95% diagnostic sensitivity and minimal complication rates (0–2%) [1,2,3,4,5]. Despite its established reliability and safety [6], EUS-TA’s diagnostic accuracy depends on factors like sampling methods, needle types, endosonographer expertise, and the availability of rapid on-site evaluation (ROSE) [3]. Integrating ROSE into EUS-TA can enhance sampling efficiency and success rates by ensuring sample adequacy, potentially boosting diagnostic accuracy by 10–15% [7,8,9,10,11].

However, conventional ROSE faces challenges like limited availability of specialized cytopathologists and high costs, necessitating methodological streamlining without compromising EUS-TA’s efficacy [12,13]. Addressing these issues, our center implemented the ROLE methodology, a novel approach where an endoscopist handles tissue sampling, smear preparation, and staining. Whitish tissue, once identified, is photographed, and these high-resolution images are immediately sent to a cytologist for real-time feedback, facilitating timely and precise pathological assessments. This method reduces the dependency on specialized personnel and resources.

ROLE’s potential to improve EUS-TA’s diagnostic precision is significant, warranting further clinical research and validation. This study aims to evaluate the ROLE protocol as an enhancement to conventional ROSE, specifically targeting the improved diagnostic performance of EUS-TA for solid pancreatic lesions.

## 2. Methods

### 2.1. Study Design and Patients

This single-center, retrospective study involved consecutive patients diagnosed with solid pancreatic lesions who underwent EUS-TA between October 2017 and March 2021. The inclusion criteria were as follows: (1) patients aged over 18 years who were initially diagnosed with solid pancreatic lesions by an abdominal enhanced electron CT or MRI, and (2) further surgery or imaging follow-up. The exclusion criteria included (1) patients with severe coagulopathy (platelet count < 50 × 10^9^/L or international normalized ratio > 1.5), and (2) the performance of EUS-TA was hindered due to factors such as the obstruction of the lesion by a vessel, an excessively long puncture distance, or a history of digestive tract diversion surgery. This study was conducted in accordance with the Declaration of Helsinki and was approved by the Institutional Ethics Committee Review Board of Peking University First Hospital.

### 2.2. Procedure of EUS-TA

All EUS-TA procedures were executed using an Olympus EU-ME2 endoscopic ultrasonography system (Olympus Ltd., Tokyo, Japan) and a GF-UCT260 curvilinear array echoendoscope (Olympus Ltd., Tokyo, Japan). To mitigate any potential technical bias, two endosonographers, each with a track record of completing at least 50 cases annually, were selected to perform the procedures in this study.

Upon ascertaining the optimal position for the echoendoscope, a needle of either a 20-gauge FNB needle (Echotip, Cook Medical Inc., Bloomington, IN, USA) or 22-gauge FNA needle (Expert TM, Boston Scientific, Marlborough, MA, USA) was employed to puncture the targeted lesions. The method of suction was selected based on lesion consistency: for harder lesions, slow-pull suction was employed, while softer lesions were subjected to suction via a 5 mL vacuum syringe. During each pass, the needle was manipulated within the lesion with a back-and-forth motion, approximately 15–20 times, to optimize specimen collection. Specimens aspirated from each EUS-TA pass were then expelled into a sterile environment using a needle stylet and a 10 mL syringe pre-filled with a saline solution.

### 2.3. Patient Grouping Based on ROLE Availability

Patients were categorized into either the ROLE group or the non-ROSE group depending on whether ROLE was employed during the EUS-TA procedure. Specimens from both groups were subsequently analyzed through histopathological and cytopathological examinations. In the non-ROLE group, standard EUS-TA was conducted, and the adequacy of each aspirated tissue strip was determined by the endoscopist’s empirical judgment. The equipment used for the ROLE method consisted of a custom-built bottom lamp, diff-quik stain (Nanjing Jiancheng, China), microscope (Olympus CX23, Tokyo, Japan), and broadcast camera (Yongnuo YN455, Shenzhen, China) equipped with cellphone connectivity and integrated with the microscope (Figure 1). Within the ROLE group, the endoscopist was responsible for immediate processing of aspiration specimens after each EUS-TA pass. This was carried out under the illumination of the custom bottom lamp. The endoscopist selectively chose whitish tissue from the aspirate for a further analysis. If no discernable whitish core tissue was identified, fluid from the needle path was used to prepare the slide for staining. The staining process involved a series of immersions: initially in a methanol fixative for 10–20 s, followed by eosin G (Stain I) for 10–15 s, and finally in thiazine dye (Stain II) for 10–15 s. After staining, the specimen was rinsed in distilled water and allowed to air-dry. The entire staining procedure was completed within an approximate time frame of 30–40 s. The stained slides were promptly assessed under the microscope to confirm the presence of cells represented.

### 2.4. Data Collection and Definition

Patients’ basic clinical characteristics, including age, sex, lesion size, lesion site (head and uncinate process, neck, body, and tail), puncture site (D1 bulb of the duodenum, D2 portion of the duodenum, stomach), boundary (clear, unclear), echo (hypoechoic, isoechoic), needle type (22G FNA, 20G FNB), and suction technique (SP, SS, SP + SS), were collected. Patients were grouped based on ROLE availability. The histological and cytological data were also collected. Following formalin fixation, the tissue strip was embedded in paraffin for subsequent sectioning. Sections were stained with Hematoxylin and Eosin (H&E) and, when applicable, additional immunostaining was performed based on the provisional diagnosis. After rinsing the needle track with saline, liquid-based cytology was conducted using centrifugation. The histological and cytological specimens were categorized into one of five diagnostic groups, malignant, suspicious, atypical, benign, or inadequate, as per established guidelines [14]. Specimens defined as malignant, suspicious for malignancy, or a specific tumor were categorized as positive. Nonspecific benign or atypical samples were considered negative. Specimens that contained inadequate material were considered false negatives. The definitive pathological diagnosis was made by an independent pathologist who was not involved in the ROLE process. Both pathologists remained blinded to each other’s diagnostic conclusions.

### 2.5. Diagnostic Criteria for Malignant and Benign Pancreatic Lesions

The gold standard for diagnoses was histological confirmation from surgically resected specimens. For non-surgical cases, diagnoses relied on the clinical course, imaging, tumor biomarkers, and treatment outcomes, confirmed by at least a 6-month follow-up. Diagnostic indicators for malignancy include the presence of one or more of the following: (1) worsening of clinical symptoms or the emergence of new symptoms, such as jaundice, obstruction, or bleeding; (2) progressive growth of the lesion or the development of distant organ metastasis, as evidenced by follow-up imaging; (3) death attributable to cancer; (4) progressive increase in tumor markers; (5) reduction in lesion size following systemic therapy or radiotherapy. Diagnostic indicators for benign conditions may include one or more of the following: (1) absence of malignancy evidence in EUS-TA; (2) improvement in or disappearance of clinical symptoms; (3) imaging findings showing stable lesions with no enlargement or metastasis after a minimum 6-month follow-up.

### 2.6. Evaluating Diagnostic Yield and Accuracy

Diagnostic yield was calculated as the percentage of patients who provided sufficient tissue or cellular material to enable a definitive pathological diagnosis. Diagnostic accuracy was defined as the proportion of FNA results that aligned with the final diagnosis. Other diagnostic performance indicators, including the sensitivity, specificity, positive predictive value (PPV), negative predictive value (NPV), procedure duration, number of needle passes, and complication rates, were also computed.

### 2.7. Statistical Analysis

Statistical analyses were conducted using SPSS version 26.0 (IBM Corp., Armonk, NY, USA). The distribution of continuous variables was evaluated using the Kolmogorov–Smirnov test. For normally distributed data, values were expressed as the mean ± standard deviation and compared between the ROLE and non-ROSE groups using Student’s *t*-test. Non-normally distributed data were presented as the median (range) and analyzed using the Mann–Whitney U-test. Categorical variables were expressed as proportions and analyzed using either the chi-squared test or Fisher’s exact test, as appropriate. In this study, diagnostic categories were divided into neoplastic (malignant tumors, suspicious) and non-neoplastic (atypical, benign, and inadequate). Cases with samples insufficient for cytological evaluation were classified as false negatives. Diagnostic accuracy, sensitivity, specificity, PPV, and NPV were calculated for the overall patients as well as for the ROLE and non-ROSE groups, respectively. A two-sided *p*-value of <0.05 was considered statistically significant.

## 3. Results

### 3.1. Baseline Characteristics

Between October 2017 and March 2021, 142 patients with solid pancreatic lesions participated in the study. However, five patients were excluded for specific reasons: two patients were found to have no lesions upon EUS examination, and three patients were lost to follow-up. Consequently, 137 patients met the inclusion criteria and were formally enrolled in the study (Figure 2). The cohort consisted of 137 patients, of which 54.0% (74 patients) were male. The mean age was 57.9 ± 12.5 years. Lesions had a mean long diameter of 3.45 ± 1.18 cm and a short diameter of 2.72 ± 1.00 cm. In the study, a 22-gauge (G) needle was most commonly used, accounting for 57.7% of patients, while a 20-G needle was utilized in the remaining 42.3% of patients. Two distinct types of needles were used in this study: FNA and FNB. Upon comparing needle size, type, and negative pressure modes, no statistically significant differences were observed between the ROLE and non-ROLE (or non-ROSE) groups, as detailed in Table 1.

### 3.2. Diagnostic Yield and Other Diagnostic Performance

The study reported an overall diagnostic yield of 93.4% (128/137). Among 137 patients subjected to EUS-TA, a significant majority, 84.7% (116/137), were diagnosed with neoplastic lesions. These included 80 ductal or mucinous epithelial tumors, 3 neuroendocrine tumors, 3 solid pseudopapillary tumors, 1 lymphopoietic system tumor, 28 tumors classified as either malignant but indistinguishable or as high-grade intraepithelial neoplasia, and 1 metastatic tumor. On the other hand, 12 patients (8.7%) were identified as non-neoplastic, and 9 patients (6.6%) had insufficient cellular material for a conclusive diagnosis. A confirmatory diagnosis for neoplastic lesions was established through surgical or imaging follow-up in 115 patients. Among these confirmatory diagnoses, 9 were true negatives, 12 were false negatives—which included samples that lacked sufficient cellular material for cytologic evaluation—and only 1 was a false positive. The study findings revealed an overall diagnostic accuracy of 90.5%, a sensitivity of 90.6%, a specificity of 90.0%, a PPV of 99.1%, and an NPV of 42.9%. The AUC for the overall patients was 81.0 (95% CI, 67.7 to 94.2). Moreover, there were 97.3% (73/75) of patients in the ROLE group who had adequate tissue samples for diagnoses, compared to 88.7% (55/62) in the non-ROSE group. In the ROLE group, the results yielded 66 true positives, 5 true negatives, 3 false negatives, and 1 false positive. Meanwhile, in the non-ROSE group, there were 49 true positives, 4 true negatives, and 9 false negatives, with no false positives recorded. Table 2 reveals significant differences between the ROLE and non-ROSE groups in terms of diagnostic yield (97.3% vs. 88.7%; *p* = 0.023), accuracy (94.7% vs. 85.5%; *p* = 0.017), and sensitivity (95.7% vs. 84.5%; *p* = 0.011). However, no statistically significant differences were observed in specificity, PPV, NPV, or AUC between the two groups (all *p*-values > 0.05; Table 2 and Figure 3A).

### 3.3. Comparative Analysis of Procedure Time, Needle Passes, and Postoperative Complications

The mean procedure time across all patients was 18.0 ± 6.5 min, with no significant difference between the ROLE and non-ROSE groups (17.9 ± 7.0 min vs. 18.3 ± 45.8 min, *p* = 0.089). However, there was a noteworthy reduction in the number of needle passes required in the ROLE group compared to the non-ROSE group (2 vs. 3, *p <* 0.001). In terms of postoperative outcomes, complications were observed in 12.6% (15/137) of the patients. These primarily included self-limiting symptoms such as nausea and abdominal pain, reported in 12 patients, and hyperthermia in 3 patients. These complications were effectively managed with intravenous antibiotic treatment and did not escalate into more severe issues like bleeding, peritonitis, or fatality. A further analysis showed no statistically significant difference in the complication rate between the ROLE and non-ROSE groups (Table 3).

### 3.4. Diagnostic Yield of FNA and FNB

Subgroup analyses were conducted to further explore the diagnostic efficacy of FNA and FNB, both with and without the use of ROLE. The findings are presented in Table 4 and Figure 3B. Notably, the diagnostic performance of FNA did not show significant differences when ROLE was incorporated, suggesting that ROLE’s added benefit might be technique-specific. On the other hand, the addition of ROLE to FNB displayed a trend toward enhanced diagnostic efficacy, although this did not reach statistical significance for most parameters. The exception was diagnostic accuracy, where FNB augmented with ROLE outperformed standalone FNB (100% vs. 93.1%, *p* = 0.025; Table 5).

## 4. Discussion

The study revealed that the ROLE protocol notably enhanced the diagnostic yield, accuracy, and sensitivity of EUS-TA for solid pancreatic lesions over traditional non-ROSE methods, with an even greater accuracy when combined with FNB. ROLE stands out as an effective method for optimizing tissue sampling, potentially reducing the need for multiple needle passes, and reliably assesses the effectiveness of punctures on targeted lesions. If a ROLE assessment is negative, endoscopists can adjust the puncture site or use alternative techniques like negative pressure or fining for improved sample collection, making ROLE a practical ROSE alternative. Contrarily, some studies, including Yang et al., found no significant improvement in EUS-FNA diagnostic rates with ROSE among highly proficient practitioners [13,15]. This could be due to experienced endoscopists’ skill in choosing puncture sites and methods, possibly diminishing ROSE’s added value. Nonetheless, for less experienced practitioners, especially during the EUS-FNA learning phase, implementing ROSE could substantially improve the diagnostic rate and accuracy.

Multiple studies have confirmed that rapid on-site evaluation (ROSE) significantly improves the quality and diagnostic yield of EUS-FNA-acquired tissue samples [7,8,9]. Additionally, ROSE, used alongside EUS-FNA, correlates strongly with final histological diagnoses in surgical patients, proving its reliability for immediate treatment decisions and reducing patient anxiety [16]. An evidence-based review recommends incorporating ROSE into the diagnostic workflow, particularly for medical centers with EUS-FNA diagnostic accuracy rates below 90% or those new to EUS techniques [17]. In our study, we implemented the ROLE technique, allowing endoscopists to selectively obtain whitish tissue for smears, while pathologists remotely and instantly evaluated specimen quality during EUS-TA. In the non-ROSE group, the decision to conclude the puncture was based on the endoscopist’s visual confirmation of the tissue strand’s integrity and their prior experience. In contrast, the ROLE group emphasizes a precise evaluation and selection of white tissue, believed to contain more diagnostically valuable components, for a cytological analysis, thereby enhancing diagnostic accuracy. This approach highlights the importance of combining experiential judgment and objective criteria to optimize diagnostic outcomes.

Several studies have investigated methods to improve or replace traditional rapid on-site evaluation (ROSE). Based on a systematic review and network meta-analysis comparing the diagnostic performance of different tissue sampling techniques for EUS-guided fine needle biopsy of pancreatic masses, the modified wet suction technique was found to offer the best performance in terms of sample integrity and adequacy, despite a higher risk of blood contamination [18]. CH-EUS-FNA outperforms conventionalEUS-FNA in diagnosing pancreatic masses, particularly in larger lesions, by offering higher sensitivity, accuracy, and sample adequacy [19]. The MOSE (Microscope-Oriented Specimen Evaluation) approach by Iwashita et al. presents an alternative method [20,21,22]. MOSE uses a stereoscopic microscope to measure whitish tissue core length in specimens, aiding tissue acquisition assessment. With a 90% concordance with histological diagnoses and a 90.7% positive predictive value, MOSE is a promising ROSE alternative, offering benefits like reduced procedural time [21,22]. Comparisons between FNB alone and FNA with ROSE show no significant differences in diagnostic success rates, with FNB at 88.1% and FNA with ROSE at 92.1% (*p* = 0.227), indicating similar tissue acquisition and diagnostic accuracy for FNB without ROSE [23,24,25]. While endoscopist-led ROSE is beneficial, a study by Savoy et al. found endoscopists to be statistically less accurate than cytopathologists in specimen quality assessment and malignancy judgments. In the ROLE approach, endoscopists identify whitish tissue, suggestive of target lesions, while cytopathologists offer precise on-line evaluations, assessing tissue quantity and potential malignancy [26,27]. Therefore, integrating ROLE into diagnostic protocols is highly recommended due to its advantages.

The integration of the ROLE approach has been observed to significantly reduce the need for multiple punctures, as evidenced in a study by Moradi A et al. [6]. This study found a notably lower number of needle passes required in the ROLE group compared to the non-ROSE group, a benefit likely due to the immediate and detailed feedback ROLE provides on the quality of tissue samples during EUS-TA procedures. Thus, ROLE minimizes unnecessary blind punctures, potentially lowering complication risks. Conversely, Kappelle and colleagues [28] achieved similar diagnostic results to ROSE by increasing needle passes, which may increase complications like abdominal pain due to potential tissue damage and inflammatory response. In our study, we did not adhere to fixed 5-7 needle passes for the non-ROSE group but relied on endoscopists’ judgment for adequate specimen collection. Notably, this did not lead to a significant uptick in complications among patients in the non-ROSE group.

While the ROLE method requires additional time for specimen processing and pathologist evaluation—usually between 5 and 9 min—it does not significantly extend the overall duration of the procedure. In fact, the adoption of ROLE minimizes the number of needle passes during EUS-TA, potentially increasing the procedure’s time efficiency. Kappelle et al.’s study showed no significant difference in the duration of EUS-FNA of lymph nodes between ROLE and non-ROSE groups (23 vs. 20 min, *p* = 0.06) [28]. However, the time taken for the pathologist’s final reading was notably shorter in the ROLE group than in the non-ROSE group (7 min and 52 s vs. 12 min and 47 s, *p <* 0.001), indicating that ROLE may also enhance efficiency for pathologists. In this study, a rare false-positive cytology case occurred in the ROLE group. Initial cytology suggested heterotypic cells, but subsequent histology was negative. It appeared that the lesion had regressed more than anticipated, showing inflammatory necrosis and mild heterotypic cell morphology, which might have affected the cytopathological diagnosis. Moreover, the endoscopist selected a less cellular sample and did not examine multiple fields during ROLE, possibly affecting lesion assessment accuracy. This study is significant for three reasons. Firstly, it establishes ROLE as a practical alternative to conventional ROSE. Endoscopists can quickly learn tissue selection, smearing, and staining within 1–2 weeks. Our microscopy setup, with a camera connected to a mobile phone, facilitates a rapid identification of cellular elements, and images are instantly sent to a remote cytopathologist for evaluation, improving EUS-TA accuracy and efficiency. This remote assessment eliminates the need for an on-site pathologist, offering faster feedback and reducing healthcare costs. Thus, ROLE has been adopted as a standard protocol for EUS-TA at our center.

This study’s strength lies in its rigorous methodology. To ensure reliability, all EUS-TA procedures were performed by two experienced endoscopists, reducing variability and operator technique biases. The rapid cytopathological assessments and final FNA histological evaluations were conducted by credentialed, independent professionals, further minimizing diagnostic influence. Additionally, those performing the ROLE technique received specialized training, ensuring competency and consistency. Despite offering valuable insights into the efficacy of the ROLE method for diagnosing solid pancreatic lesions, the study’s limitations are noteworthy. Its retrospective cohort design inherently risks introducing data collection and patient selection biases. Variations in puncture techniques, suction methods, and needle types could have influenced selection biases. However, a thorough comparison between the ROLE and non-ROSE groups showed no significant differences, somewhat alleviating these concerns. In addition, we explore the contrasting results between an RCT, which found no significant benefit of rapid on-site evaluation (ROSE) in conjunction with fine needle biopsy (FNB) for pancreatic tumors, and our subgroup analysis showing improved diagnostic accuracy with the combination of FNB and rapid on-site evaluation (ROLE). This discrepancy may be attributed to several factors, including the superior tissue volume and quality obtained through FNB, the potential for selection bias in a small sample size, and the precision required in needle placement. These findings underscore the importance of further prospective studies to rigorously evaluate the utility of ROLE in enhancing the diagnostic accuracy of FNB for pancreatic tumors, considering the limitations observed in previous research.

In conclusion, this study potentially demonstrates the effectiveness of the ROLE protocol in enhancing the diagnostic performance of EUS-TA for solid pancreatic lesions. The ROLE method not only equates to the diagnostic results achieved by conventional non-ROSE methods but also presents benefits such as fewer needle passes and no substantial increase in overall procedure duration. These results suggest that ROLE can optimize diagnostic processes without sacrificing diagnostic accuracy, thereby enhancing clinical efficiency and patient experience.

## Figures and Tables

**Figure 1 diagnostics-14-00597-f001:**
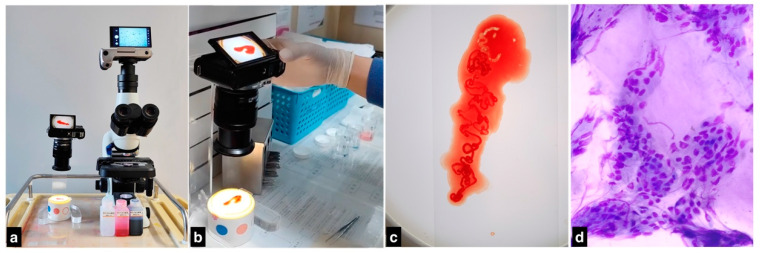
Protocol of ROLE methodology. (**a**) Equipment setup: the ROLE equipment comprises a bottom lamp, diff-quik stain, and microscope integrated with a broadcast camera featuring cellphone functionality. (**b**,**c**) Sample processing: Whitish tissue specimens were selectively chosen from the aspirate. A cellular smear was prepared and subjected to diff-quik staining. (**d**) Cell evaluation and feedback: The stained slide was inspected under the microscope to verify the presence of representative cells. Photomicrographs were captured and promptly sent to a cytopathologist for evaluation via WeChat.

**Figure 2 diagnostics-14-00597-f002:**
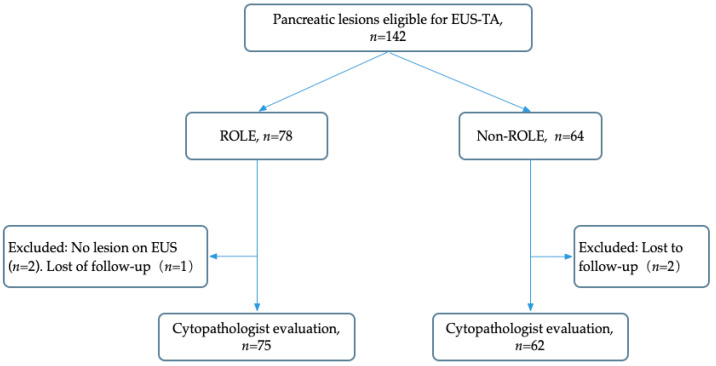
Study flow diagram.

**Figure 3 diagnostics-14-00597-f003:**
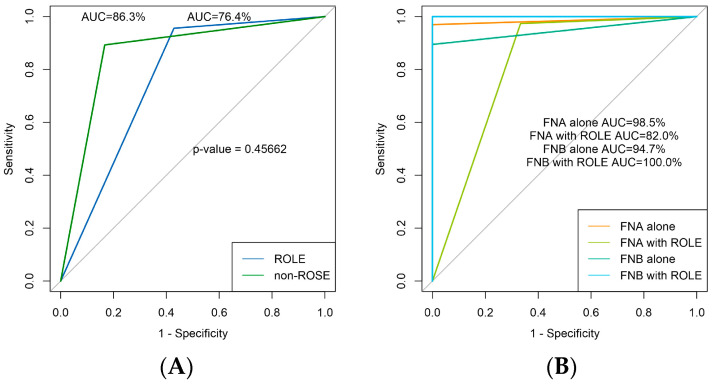
Receiver Operating Characteristic (ROC) analysis comparing diagnostic parameters. (**A**) ROC curves for the ROLE and non-ROSE group. (**B**) ROC curves for FNA alone, FNA with ROLE, FNB alone, and FNB with ROLE group; AUC: Area Under the Curve.

**Table 1 diagnostics-14-00597-t001:** Basic clinical characteristics.

Clinical Characteristics	Total (*n* = 137)	ROLE (*n* = 75)	Non-ROSE (*n* = 62)	*p*-Value
Age, years, mean ± SD	57.9 ± 12.5	58.5 ± 13.0	57.3 ± 12.0	0.559
Sex, male, *n* (%)	74 (54.0)	37 (49.3)	37 (59.7)	0.227
Lesion size (cm, mean ± SD)				
Long diameter	3.45 ± 1.18	3.46 ± 1.12	3.43 ± 1.27	0.894
Short diameter	2.72 ± 1.00	2.79 ± 1.07	2.63 ± 0.91	0.328
Lesion site, *n* (%)				0.318
Head and uncinate process	90 (65.7)	53 (70.7)	37 (59.7)	
Neck	21 (15.3)	11 (14.7)	10 (16.1)	
Body and tail	26 (19.0)	7 (13.7)	15 (24.2)	
Puncture site, *n* (%)				
D1	95 (69.3)	58 (77.3)	37 (59.7)	0.083
D2	12 (8.8)	5 (6.7)	7 (11.3)	
Stomach	30 (21.9)	12 (16.0)	18 (29.0)	
Boundary, *n* (%)				
Clear	30 (21.9)	17 (22.7)	13 (21.0)	0.811
Unclear	107 (78.1)	58 (77.3)	49 (79.0)	
Echo, *n* (%)				
Hypoechoic	126 (92.0)	69 (92.0)	57 (91.9)	1.000
Isoechoic	11 (8.0)	6 (8.0)	5 (8.1)	
Needle type, *n* (%)				
22G FNA	79 (57.7)	41 (54.7)	38(61.3)	0.435
20G FNB	58 (42.3)	34 (45.3)	24 (38.7)	
Suction technique, *n* (%)				
SP	67 (48.2)	39 (52.0)	28 (43.8)	0.163
SS	18 (12.9)	12 (16.0)	6 (9.4)	
SP + SS	54 (38.8)	30 (32.0)	24 (6.9)	

ROLE: rapid on-line evaluation; ROSE: rapid on-site evaluation; FNA: fine needle aspiration; FNB: fine needle biopsy; SP: slow-pull suction; SS: standard suction; SD: standard deviation.

**Table 2 diagnostics-14-00597-t002:** Comparison of diagnostic performance between ROLE group and non-ROSE group.

Diagnostic Performance	Total (*n* = 137)	ROLE Group (*n* = 75)	Non-ROSE Group (*n* = 62)	*p*-Value
Diagnostic yield, % (95% CI)	93.4 (87.9 to 97.0)	97.3 (90.7 to 99.7)	88.7 (78.1 to 95.3)	0.023
Accuracy, % (95% CI)	90.5 (84.3 to 94.9)	94.7 (86.9 to 98.5)	85.5 (74.2 to 93.1)	0.027
Sensitivity, % (95% CI)	90.6 (84.1 to 95.0)	95.7 (87.8 to 99.1)	84.5 (72.6 to 92.7)	0.011
Specificity, % (95% CI)	90.0 (55.5 to 99.7)	83.3 (35.9 to 99.6)	100.0 (39.8 to 100.0)	1
PPV, % (95% CI)	99.1 (95.3 to 100.0)	98.5 (92.0 to 100.0)	100 (92.7 to 100.0)	1
NPV, % (95% CI)	42.9 (21.8 to 66.0)	62.5 (24.5 to 91.5)	30.8 (9.1 to 61.4)	0.094
AUC (95% CI)	81.0 (67.7 to 94.2)	76.4 (56.4 to 96.3)	86.3 (69.5 to 100.0)	0.457

ROLE: rapid on-line evaluation; ROSE: rapid on-site evaluation; PPV: positive predictive value; NPV: negative predictive value; AUC: Area Under the Curve.

**Table 3 diagnostics-14-00597-t003:** Comparison of procedure time, number of needle passes, and complications between ROLE group and non-ROSE group.

Outcomes	Total (*n* = 137)	ROLE Group (*n* = 75)	Non-ROSE Group (*n* = 62)	*p*-Value
Procedure time (min, mean ± SD)	18.0 ± 6.5	17.9 ± 7.0	18.3 ± 5.8	0.716
Number of needle passes, *n* (%)				
1	5 (3.6)	4 (5.3)	1 (1.6)	<0.001
2	50 (36.5)	39 (52.0)	11 (17.7)	
3	48 (35.0)	24 (32.0)	24 (38.7)	
4	30 (21.9)	7 (9.3)	23 (37.1)	
5	4 (2.9)	1 (1.3)	3 (4.9)	
Adverse events, *n* (%)				
None	122 (89.1)	70 (93.3)	52 (83.9)	0.210
Abdominal discomfort	12 (8.8)	4 (5.3)	8 (12.9)	
Fever	3 (2.2)	1 (1.3)	2 (3.2)	

ROLE: rapid on-line evaluation; ROSE: rapid on-site evaluation; SD: standard deviation. *p*-Values indicate the level of statistical significance between ROLE and non-ROSE group.

**Table 4 diagnostics-14-00597-t004:** Comparative Diagnostic performance for FNA and FNB, both with and without ROLE.

Diagnostic Performance	FNA Alone	FNA with ROLE	FNB Alone	FNB with ROLE
Diagnostic yield, % (95% CI)	91.1 (82.6 to 96.4)	95.1 (83.5 to 99.4)	96.6 (88.1 to 99.6)	100.0 (100.0 to 100.0)
Accuracy, % (95% CI)	88.6 (79.5 to 94.7)	90.2 (76.9 to 97.3)	93.1 (83.3 to 98.1)	100.0 (89.7 to 100.0)
Sensitivity, % (95% CI)	97.2 (90.2 to 99.7)	97.4 (86.2 to 99.9)	95.8 (85.7 to 99.5)	100.0 (88.1 to 100.0)
Specificity, % (95% CI)	87.5 (47.3 to 99.7)	66.7 (9.4 to 99.2)	100.0 (69.2 to 100.0)	100.0 (47.8 to 100.0)
PPV, % (95% CI)	98.6 (92.3 to 100.0)	97.4 (86.2 to 99.9)	100.0 (92.3 to 100.0)	100.0 (88.1 to 100.0)
NPV, % (95% CI)	77.8 (40.0 to 97.2)	66.7 (9.4 to 99.2)	83.3 (51.6 to 97.9)	100.0 (47.8 to 100.0)
AUC (95% CI)	98.5 (95.5 to 100.0)	82.0 (49.3 to 100.0)	94.7 (87.7 to 100.0)	100.0

ROLE, rapid on-line evaluation; FNA, fine needle aspiration; FNB, fine needle biopsy; PPV, positive predictive value; NPV, negative predictive value; AUC: Area Under the Curve.

**Table 5 diagnostics-14-00597-t005:** *p*-Values for comparisons of diagnostic performance between FNA and FNB, with and without ROLE.

	*p*-Value					
	FNA vs. FNA + ROLE	FNA vs. FNB	FNA vs. FNB + ROLE	FNA + ROLE vs. FNB	FNA + ROLE vs. FNB + ROLE	FNB vs. FNB + ROLE
Diagnostic yield (%)	0.252	0.206	0.055	1.000	0.498	0.167
Accuracy (%)	0.731	0.375	0.100	0.715	0.121	0.025
Sensitivity (%)	1.000	0.480	1.000	1.000	1.000	0.152
Specificity (%)	0.375	0.231	1.000	0.231	0.375	-
PPV (%)	1.000	1.000	1.000	0.452	1.000	-
NPV (%)	1.000	1.000	0.505	0.117	0.375	0.470
AUC	0.332	0.347	0.324	0.461	0.289	0.159

ROLE, rapid on-line evaluation; FNA, fine needle aspiration; FNB, fine needle biopsy; PPV, positive predictive value; NPV, negative predictive value; AUC: Area Under the Curve.

## Data Availability

The data presented in this study are available in the article.

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
