# Peer review of "A Rapid On-Line Evaluation (ROLE) Protocol in the Diagnostic Performance Improvement in Endoscopic Ultrasound-Guided Tissue Acquisition for Solid Pancreatic Lesions"

_diagnostics, 2024, doi:10.3390/diagnostics14060597_

Round 1

Reviewer 1 Report

Comments and Suggestions for Authors

Very interesting study on an important topic.

The authors should specify which needles they used for tissue sampling. Did they use newer end-cutting FNB needles? THey mentioned only the size of the needle. They should comment more also on the sampling techniques used (for example slow pull, wet suction, and so on....)

In this regard, the authors should cite the latest advancements in this field, mentioning the most important relevant studies (cite PMID: 33481633; PMID: 36657607 )

The authors should comment more on the handling of the sample collected. How did the pathologist analyze it?

The timespan to assess the clinical course of the disease (gold standard for accuracy analysis) should be 1 year, not just 6 months

Author Response

thank you for your comments and suggestion, Please see my response in the attachment!

TRANSLATE with x English

Arabic Hebrew Polish
Bulgarian Hindi Portuguese
Catalan Hmong Daw Romanian
Chinese Simplified Hungarian Russian
Chinese Traditional Indonesian Slovak
Czech Italian Slovenian
Danish Japanese Spanish
Dutch Klingon Swedish
English Korean Thai
Estonian Latvian Turkish
Finnish Lithuanian Ukrainian
French Malay Urdu
German Maltese Vietnamese
Greek Norwegian Welsh
Haitian Creole Persian  

TRANSLATE with COPY THE URL BELOW Back EMBED THE SNIPPET BELOW IN YOUR SITE Enable collaborative features and customize widget: Bing Webmaster Portal Back

TRANSLATE with x English

Arabic Hebrew Polish
Bulgarian Hindi Portuguese
Catalan Hmong Daw Romanian
Chinese Simplified Hungarian Russian
Chinese Traditional Indonesian Slovak
Czech Italian Slovenian
Danish Japanese Spanish
Dutch Klingon Swedish
English Korean Thai
Estonian Latvian Turkish
Finnish Lithuanian Ukrainian
French Malay Urdu
German Maltese Vietnamese
Greek Norwegian Welsh
Haitian Creole Persian  

TRANSLATE with COPY THE URL BELOW Back EMBED THE SNIPPET BELOW IN YOUR SITE Enable collaborative features and customize widget: Bing Webmaster Portal Back

Reviewer 2 Report

Comments and Suggestions for Authors

I would like to thank you for the opportunity to review this article. The authors demonstrate that remotely diagnosed rapid on-line evaluation (ROLE) is a useful substitute for EUS-FNA for pancreatic tumors when a cytologist is not available. This article is thought-provoking for physicians performing EUS-FNA. However, there are a few comments that should be considered upon acceptance and are listed below.

1.     I understand that this is a retrospective study, however, how was the indication for performing ROLE determined in the beginning? Please describe in the method.

2.     The results of an RCT using the FNB needle for pancreatic tumors are available. Since there was no difference in the positive diagnosis rate with or without rapid on-site evaluation (ROSE) and the time to perform ROSE was longer in the ROSE group, the article concludes that ROSE is not recommended and is cited in the article (Ref. No. 13). The results of the present subgroup analysis show that the diagnostic accuracy of the combination of FNB and ROLE was better than that of FNB alone. What could be the reason for the different results?

3.     What are the criteria for determining the end of puncture in the non-ROSE group?I do not understand why the diagnostic accuracy of ROLE is higher when the endoscopists themselves are checking the white tissue obtained by FNA. Please discuss this point in your discussion.

4.     How is the cost to set up this online diagnostic system?

Author Response

Thank you for your comments and suggestions! please see my response in the attachment.

TRANSLATE with x English

Arabic Hebrew Polish
Bulgarian Hindi Portuguese
Catalan Hmong Daw Romanian
Chinese Simplified Hungarian Russian
Chinese Traditional Indonesian Slovak
Czech Italian Slovenian
Danish Japanese Spanish
Dutch Klingon Swedish
English Korean Thai
Estonian Latvian Turkish
Finnish Lithuanian Ukrainian
French Malay Urdu
German Maltese Vietnamese
Greek Norwegian Welsh
Haitian Creole Persian  

TRANSLATE with COPY THE URL BELOW Back EMBED THE SNIPPET BELOW IN YOUR SITE Enable collaborative features and customize widget: Bing Webmaster Portal Back

TRANSLATE with x English

Arabic Hebrew Polish
Bulgarian Hindi Portuguese
Catalan Hmong Daw Romanian
Chinese Simplified Hungarian Russian
Chinese Traditional Indonesian Slovak
Czech Italian Slovenian
Danish Japanese Spanish
Dutch Klingon Swedish
English Korean Thai
Estonian Latvian Turkish
Finnish Lithuanian Ukrainian
French Malay Urdu
German Maltese Vietnamese
Greek Norwegian Welsh
Haitian Creole Persian  

TRANSLATE with COPY THE URL BELOW Back EMBED THE SNIPPET BELOW IN YOUR SITE Enable collaborative features and customize widget: Bing Webmaster Portal Back

Round 2

Reviewer 1 Report

Comments and Suggestions for Authors

The revised version of the paper is OK. THank you!

Reviewer 2 Report

Comments and Suggestions for Authors

The manuscript has been revised well. I think this manuscript will be acceptable.